# Optimum Mix Design for 3D Concrete Printing Using Mining Tailings: A Case Study in Spain

**Martina-Inmaculada Álvarez-Fernández [1,\*], María-Belén Prendes-Gero [2], Celestino González-Nicieza [1], Diego-José Guerrero-Miguel [1] and Juan Enrique Martínez-Martínez [2]**

1   Mining Exploitation Department, EIMEM, University of Oviedo, 33003 Oviedo, Spain;
    celestino@dinrock-uniovi.com (C.G.-N.); diego@dinrock-uniovi.com (D.-J.G.-M.)
2   Construction Department, EPM, University of Oviedo, 33003 Oviedo, Spain;
    belen@dinrock-uniovi.com (M.-B.P.-G.); quique@constru.uniovi.es (J.E.M.-M.)
\*   Correspondence: inma@uniovi.es

**Abstract:** A mix design, using a mixture of sand and mine tailings as aggregates, was selected to produce a cement-based 3D printing material suitable for building purposes. Different dosage rates of mine tailings, water, superplasticizers, and accelerators were added to the mixture with the end of looking for the optimum strength, workability and buildability. The term buildability includes aspects such as pumpability and printability. Different tests were carried out in order to compare homogeneous material strength with printed material strength, to evaluate the bonding strength between filaments, and to establish the relationship between fresh behaviour and buildability for printing applications. Finally, a mixture with 20% of recycled materials demonstrated its ability to be used as concrete printing material in the construction industry in the frame of circular economy concept.

**Keywords:** concrete printing; printing buildability; rheological properties; fresh concrete

## 1. Introduction

The mining tailings generated in the processes of treatment and concentration of ores are defined as solid mineral tailing with a reduced grain size, between silt (4 μm to 62 μm) and sand (62 μm and 4 mm). These mining tailings are produced, transported, and deposited in the form of sludge in different structures or dumped into the aquatic environment. These deposits present a series of problems, especially when mining activity ceases, such as land occupation; the need to guarantee impermeability to prevent the escape of contaminants; and the control of seismic risks, erosion, piping, overflows, foundation failures, etc., to guarantee a certain long-term stability and minimise the risk of massive mobilisation of contaminants.

In the last 80 years there are more than 120 relevant failures documented [1,2]. The most recent incident occurred on 25 January 2019, at the Brumadinho dam in Brazil, causing 250 deaths and 20 missing persons (counted 670 days after the event) [3].

Therefore, if mining tailings are not properly managed, these tailings can cause irreversible damage to the environment and constitute a danger to humans. Despite the environmental challenges associated with mining and its tailings, mining industries can be integrated to form a model of a circular economy that promotes the reduction of tailings through recycling and reuse of these materials.

One alternative that is gaining strength is the reuse of these materials as a partial replacement for the aggregate used in concrete and mortar. There are some experimental studies on the use of marble tailing [4,5] or even the floating of similar minerals partially replacing the volume of aggregates when making concretes or mortars.

Esmaeli and Aslani [6] analysed the use of copper mine tailing in concrete and revealed the success of the partial replacement material for the cement. Gou et al. [7] reviewed the

potential utilization of tailings as a replacement for fine aggregates, such as supplementary cementitious materials (SCMs) in mortar or concrete and in the production of cement clinker. Ince [8] reused gold-mine tailings in cement mortars, showing an improvement of properties such as compressive strength, water penetration depth, porosity.

However, in these studies the cement or water content of the mixtures was not considered to be a critical parameter, seeing as neither the pumpability nor the buildability of the mixtures were being pursued.

The application proposed in this research is the possibility of reusing mining tailings as a raw material in a cementitious mixture that can be used in an additive manufacturing process of structural elements.

As an additive manufacturing process, the 3D concrete printing builds concrete components dependent on an additive, layer-based manufacturing technique [9,10]. An important added value is that this method can be used to build complex geometries without formwork.

Fresh properties for printing material were evaluated through rheology tests by Le et al. [11]. The extrudability was evaluated with 9 mm wide filaments (printed from a 9 mm nozzle), and each filament was 300 mm long. However, this width is very small for most applications. The research now presented is more aligned with [12], which proposed the use of more friendly tests, such as slump or slump-flow, and that the buildability be evaluated in terms of maximum height printed before collapsing.

Other research such as [13,14] are good reviews about the state of technology and the concrete mixtures' properties. According to [15], the major challenge in concrete printing is to identify and maintain the mixture characteristics suitable for both printing and stacking up in layers. In this study, fresh mixtures with silica fume and superplasticizer were characterized for printability based on their rheological properties. Mix proportion and fresh properties of fly ash-based geopolymer for 3D concrete printing were developed by Panda and Tan [16].

According to the bibliography, the use of recycled materials from mining processes has not yet been considered for this demanding application of additive manufacturing. In fact, a consistency must be achieved that allows the fresh material to flow, be pumpable, and adapt to complex shapes. At the same time, a buildability must be achieved that allows some strands to be deposited on top of others with an acceptable deformation in a time that is appropriate to the construction speed. Moreover, the material must be suitable to provide a good union between layers.

The end of this paper is to demonstrate the employability of mining tailings from a flotation process as materials that partially replace conventional aggregate and to manufacture a mixture with the characteristics of strength, consistency, workability, and buildability required of a printable material. These are properties which, in the case of additive manufacturing, are opposed because, for example, a good workability normally implies a bad consistency and vice versa.

## 2. Research Objectives and Process

The main objective of this study is to determine the optimal content of recycled material of mining origin that can be introduced into a cementitious mix so that its strength and workability are compatible for additive 3D manufacturing.

In addition, it is intended to verify the viability of evaluating the suitability of the material from the point of view of its pumpability and extrudability by means of simple tests that are easily carried out on site, such as the Abrams mini cone [17] or the flow table test [18].

To characterize the adhesion between layers, a new test is proposed: loading with a chisel [19], which requires simple equipment, specifically a universal press and a chisel-type tool, which would be much simpler than, for example, a direct cut test.

## 3. Materials and Test Methods

The aim is to obtain an optimised mixture from recycled materials (mining tailings) which, together with conventional aggregates, cement, water and the necessary additives, presents characteristics of strength, workability and buildability adapted to additive manufacturing.

### 3.1. Materials

The tailings come from one abandoned deposit in the north of Spain. From a granulometric point of view, they are a fine sand, as more than 99% is less than 0.25 mm, with an average size of 0.1 mm and the content of fines (silts and clays) of 18%. The coefficient of uniformity, Cu, is 2.40, and therefore, they are a very uniform material.

The specific weight of the solid particles is 26.1 kN/m$^3$. Mineralogically, the tailings are mainly made up of silica (80%), calcite (15%), and feldspars (5%).

The filler aggregate is a commercial sand, also of siliceous origin, whose granulometric curve can be seen in Figure 1. With regard to the cement used, it is type IV/B (V) 32.5 N Portland cement. In some cases, to optimise the mixture, the addition of a superplasticizer type MasterRheobuild-1000 and/or an accelerator type MasterSeed has been tested. Both of them are products of MBCC Group [20].

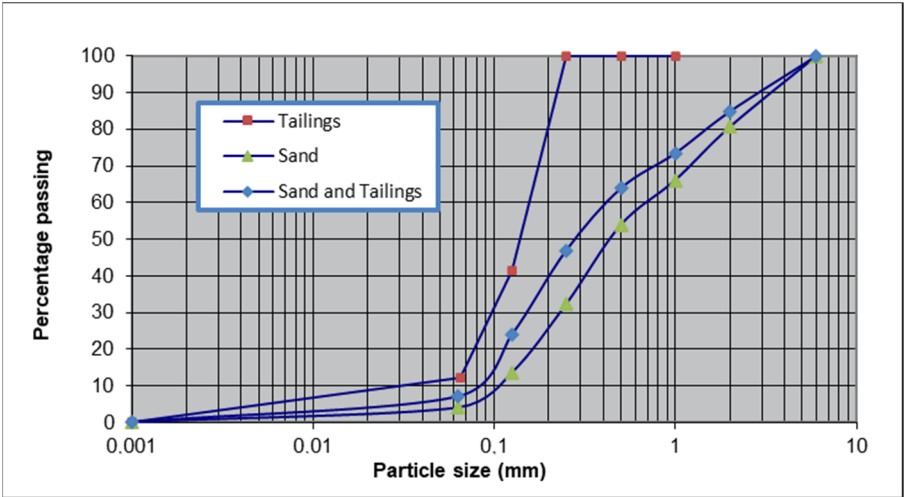

**Figure 1.** Granulometric curve of the tailings, the sand, and the final mixture of sand and tailings.

### 3.2. Test Methods

The first step was to establish a common mixture and kneading methodology for all tested mixtures. Special attention was paid to ensure that the water temperature and kneading times were always the same due to the influence of these two parameters [21,22]. The summary of this process is shown in Table 1.

**Table 1.** Summary of the kneading process.

| Time Rate (min) | Description |
| --- | --- |
| 0–1 | The tailings are mixed with 45% of the water |
| 1–2 | The rest of the aggregate (sand) is added |
| 2–3 | The cement is added |
| 3–4 | The remaining water and additives are added |
| 4–7.5 | The final mixture is carried out at maximum speed |

The test was carried out in four successive phases as described below, that attempt to resolve the conditioning factors of the material to be designed: strength, consistency, workability, and buildability. In the fourth phase, new techniques were developed to predict buildability from the fresh strength of cementitious mixtures. For the kneading and

taking into account the size of the granulometry of the materials, a double sigma mixer was used.

- Phase 1: The water/cement ratios and the content of recycled tailings were adjusted to obtain a simple compressive strength of the material close to 25 MPa at 28 days (Table 2). This phase is necessary given that the granulometry of the tailings is very fine, which requires a higher moisture content so that the mixture has a suitable consistency for pumping and deposition, although this penalises the strength. During this phase, the samples are subjected to simple compressive and indirect tensile strength tests according to the UNE-EN:12390-3:2020 Standard [23].

- Phase 2: The dosages were optimised by adding a superplasticizer that improved workability by reducing the water content (which improves strength behaviour) (Table 3). Workability was evaluated by means of mini-slump tests and flow table tests. Conventional methods were chosen because the granulometry of the material and because they are easier to implement on a construction site.

- Phase 3: The buildability of the mixtures was tested by fabrication. Samples made by depositing filaments on top of each other were tested to corroborate workability and evaluate the adhesion between layers. This adherence was evaluated by means of a penetration test with a metallic chisel, comparing it with that of the homogeneous material, without discontinuities between layers. This test has already been used for this purpose on materials with joints, such as slates.

- Phase 4: A simple methodology was proposed to characterise the rheology of this type of material in order to evaluate how it improves its strength in the fresh state and therefore its capacity to withstand the addition of new filaments. These techniques were applied to the two mixtures that had shown the best buildability behaviour.

**Table 2.** Composition of the mixtures tested in phase 1.

| ID | Aggregate | | Cement (kg) | Cement/ Aggregate Ratio | Water (L) | Water/ Cement Ratio | Compressive Strength (UCS) (MPa) | Standard Deviation of UCS | Tensile Strength (TS) (MPa) | Standard Deviation of TS |
|---|---|---|---|---|---|---|---|---|---|---|
| | Tailings (kg) | Sand (kg) | | | | | | | | |
| **F1-M1** | 59 | - | 12 | 0.2 | 29 | 2.50 | 0.9 | 0.03 | 0.3 | 0.02 |
| **F1-M2** | 67 | - | 13 | 0.2 | 20 | 1.50 | 3.3 | 0.31 | 0.8 | 0.08 |
| **F1-M3** | 59 | - | 24 | 0.4 | 18 | 0.75 | 9.3 | 0.45 | 1.3 | 0.15 |
| **F1-M4** | 24 | 36 | 24 | 0.4 | 17 | 0.70 | 13.4 | 0.51 | 1.8 | 0.30 |
| **F1-M5** | 12 | 48 | 24 | 0.4 | 15 | 0.63 | 23.5 | 0.40 | 2.9 | 0.20 |

**Table 3.** Composition of the mixtures tested in the Phase 2.

| ID | Water Content (%) | Cement/Aggregate Ratio | Water/Cement Ratio | Superplasticizer Content (%) | Other Additives | Compressive Strength (UCS) (MPa) | Standard Deviation of UCS |
|---|---|---|---|---|---|---|---|
| **F2-M1** | 21.0 | 0.4 | 0.50 | 0.5 | NO | 26.3 | 0.99 |
| **F2-M2** | 18.5 | 0.4 | 0.46 | 1.0 | NO | 28.7 | 1.02 |
| **F2-M3** | 18.5 | 0.4 | 0.46 | 1.0 | Accelerator | 26.6 | 0.91 |
| **F2-M4** | 18.5 | 0.4 | 0.46 | 0.5 | NO | 27.2 | 1.67 |

The mini-slump test is a simple and fast method to study the consistency of cement paste. The test was originally developed by Kantro [24] and later modified by Zhor and Bremner [25]. The mini-slump cone used has a top diameter of 70 mm, a bottom diameter of 92 mm, and a height of 120 mm (Figure 2). The cone is lifted, and the average spread of the paste, measured along two diagonals and two medians, is recorded after one minute.

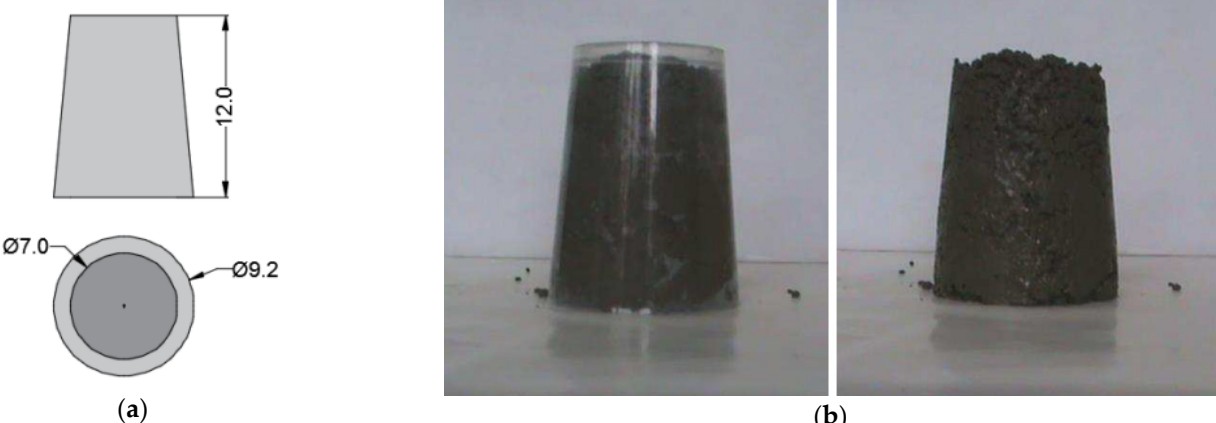

**Figure 2.** Mini-slump test: (**a**) cone geometry (cm); (**b**) aspect of one test.

This test has been completed with the flow table test [18]. To perform the test, a cone mould with a top diameter of 13 cm and a bottom diameter of 20 cm is placed in the centre of one square plate with 70 cm sides. The cone is filled in two layers, each of which is compacted with a tamping filament. The plate is lifted with one attached handle a distance of 40 mm and then dropped a total of 15 times. The horizontal spread of the concrete is measured. In this case the same mould was used in the mini-slump test.

## 4. Results

The main results of the four phases of the test are described below.

### 4.1. Phase 1: Determination of the Maximum Content of Tailings in the Mixture

Different mixtures were created, optimising the content of tailings in the aggregate and the amount of water, so that all the mixtures were workable. For all of them, the strength to simple compressive and to indirect tensile at 14 and 28 days were evaluated through at least three tests for each mixture. Although this number of tests is less than the standard says, the goal of the study is to look for the optimised mixture and then increase the number of test with the idea of industrial implementation.

Of the checked mixtures, many of them turned out to be unworkable, while the five most significant are shown in Table 2. In it, the results are the average value from three samples of each mixture. Attempts were made to work with the maximum quantity of recycled aggregates (100%), given that the first three mixtures do not contain sand. However, they require very high humidities that result in low values of simple compressive strength (below 10 MPa). The incorporation of sand (the last two mixtures) improves strength, exceeding 23 MPa for 20% recycled aggregate (F1-M5). This is mainly due to the fact that the incorporation of more sands makes it possible to reduce the water/cement ratio. This same dosage, in spite of the fact that its simple compressive strength does not reach the values required for structural concrete, will be taken as the starting point for phase 2.

### 4.2. Phase 2: Consistency and Workability

In order to not reduce the content of recycled aggregates in the final mixture, it was decided to add a superplasticizer to the material to reduce the water content while maintaining workability and thus improving the strength of the mixture.

In this way, four new mixtures were prepared. The characteristics of them are shown in Table 3. All of them have the same composition as the F1-M5 mixture, with the exception of the water content and the additives. The results are the average value from three samples of each mixture. As can be seen, in all cases, the reduction in the water/cement ratio makes it possible to achieve high simple compressive strengths above 25 MPa that represented the first objective.

The F2-M3 mixture aims to improve the properties of the F2-M2 mixture for 3D printing applications by adding an accelerator so that the filaments reach the necessary strength earlier to allow their successive overlapping, one on top of the other.

The workability and consistency of these materials was evaluated, firstly, by means of a test with the Abrams mini cone [17]. To define the time during which the material maintains its consistency characteristics, three tests were carried out for each mixture: one with the fresh material, another after 10 min, and another 20 min after mixing. Figure 3 shows the photographs of the four mixtures at the three times. The results of these tests are shown in Table 4. In it, D1 and D2 are the top and bottom diameter of the cone after the test.

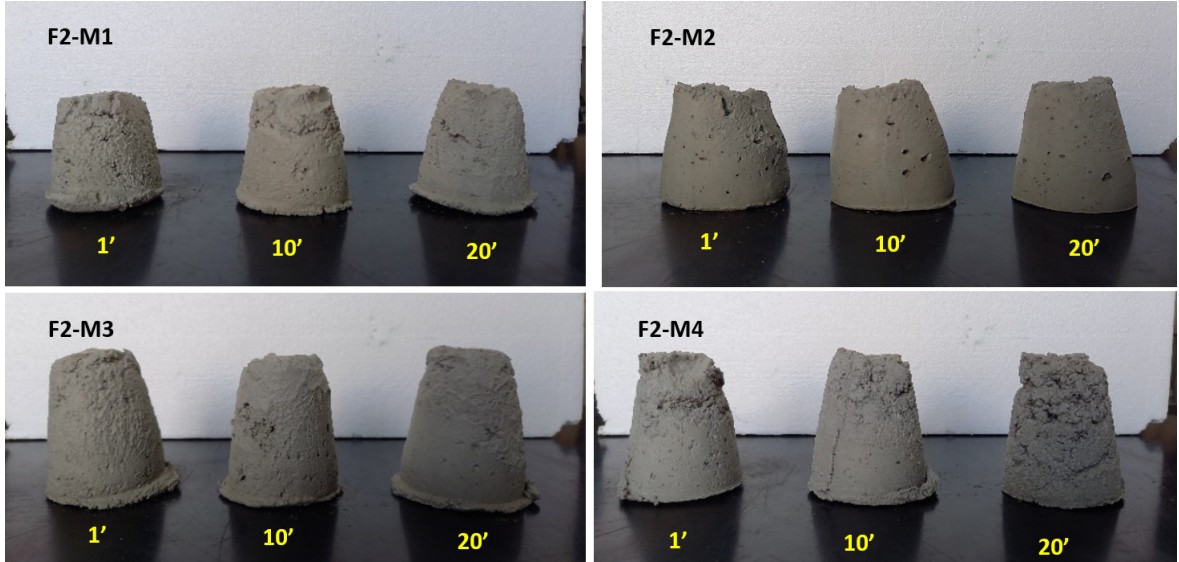

**Figure 3.** Mini-slump tests.

**Table 4.** Results of the test with Abrams mini cone in the material of the filaments.

| ID | Time after Mixing (min) | D1 × D2 (mm) |
|---|---|---|
| | 1 | 78 × 78 |
| F2-M1 | 10 | 78 × 78 |
| | 20 | 78 × 75 |
| | 1 | 70 × 86 |
| F2-M2 | 10 | 75 × 92 |
| | 20 | 74 × 85 |
| | 1 | 74 × 76 |
| F2-M3 | 10 | 73 × 77 |
| | 20 | 73 × 75 |
| | 1 | 71 × 76 |
| F2-M4 | 10 | 71 × 75 |
| | 20 | 73 × 75 |

Taking into account that the top and bottom diameters are initially of 70 and 92 mm (Figure 2), it is possible to say that the F2-M4 sample presents the smallest deformation due to its water and superplasticizer content, and with the same superplasticizer content but smaller water content, F2-M1 presents the biggest deformation. On the other hand, F2-M2 and F2-M3, with the same water content that F2-M4 but with more superplasticizer content, present intermediate deformation, although F2-M3 stabilizes first due to its higher accelerator content.

Figure 4 shows the look of the four mixtures after being tested in the flow table apparatus [18]. The final dimensions, after the test, are shown in Table 5. In it, D1 and

D2 are the top and bottom diameter of the cone after the test that initially had a value of 13 and 20 cm respectively.

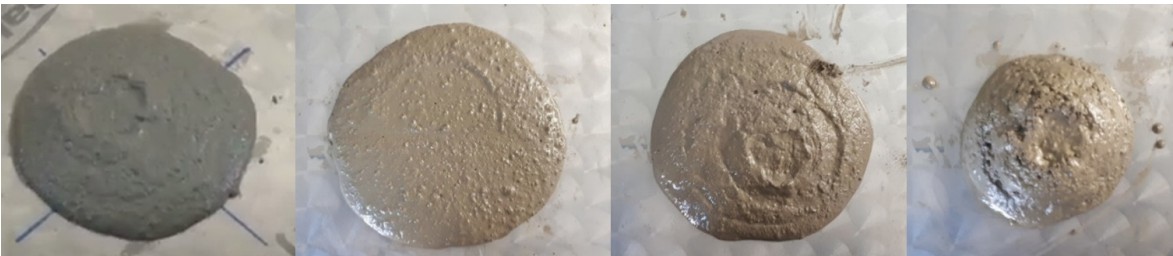

**Figure 4.** Look of the mixtures after the flow table tests.

**Table 5.** Flow table tests results.

| ID | Time after Mixing (min) | D1 × D2 (cm) |
|---|---|---|
| F2-M1 | 1 | 19.0 × 19.0 |
| | 10 | 18.5 × 18.5 |
| | 20 | 18.5 × 17.0 |
| F2-M2 | 1 | 24.0 × 24.0 |
| | 10 | 21.5 × 23.0 |
| | 20 | 20.5 × 22.0 |
| F2-M3 | 1 | 19.0 × 19.0 |
| | 10 | 20.0 × 21.0 |
| | 20 | 19.5 × 20.0 |
| F2-M4 | 1 | 15.5 × 16.0 |
| | 10 | 13.5 × 13.5 |
| | 20 | 12.0 × 12.5 |

As in the previous test, F2-M4 presents the smallest deformation while F2-M1 presents the biggest deformation, and F2-M3 with intermediate deformations presents, with the time, less variation in the diameter due to its higher accelerator content.

*4.3. Phase 3: Buildability*

With phase 2 formulations, that is to say with F2-M1, F2-M2, F2-M3, and F2-M4 mixtures, small-scale filament deposition tests are initiated. Along the "printing" process with a 3D robot, filaments are overlapped on top of each other until the desired piece height is reached.

In order to try to evaluate the behaviour of the material and validate its use before starting an industrial process, a device was designed to reproduce this type of printing on a laboratory scale. For this purpose, a carriage guided by rails was constructed to simulate the pumping of the filaments and ensured that they overlapped correctly (see Figure 5). The device contained graduated marks to determine the distance between the pipe mouth and the ground (height of the filament) and a screw fastening system to vary this distance and thus to superimpose several filaments. At the end of the pipe where the product was pumped (diameter 50 mm), a curved nozzle was placed at a 45° angle, which allowed for cleaner deposition, leaving one horizontal surface without dragging. The pump was a laboratory gear pump with a flow rate of 12 L/min and a power of 0.3 kW.

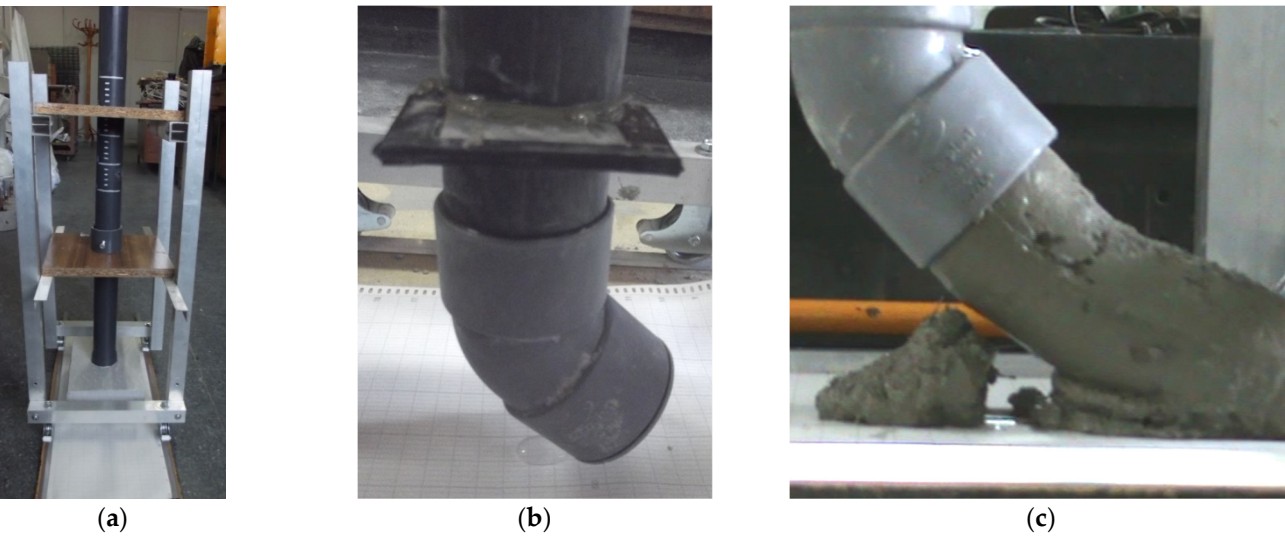

**Figure 5.** Printing device in the laboratory: (**a**) printing carriage; (**b**) curved nozzle, and (**c**) filament deposition.

With this system, samples were prepared by adding filaments, and the behaviour of these ones as they overlap was checked. The common effect in all cases was the crushing of the lower layers as a result of the weight of the new filaments, although in all cases the consistency allows the addition of up to five layers. Figure 6 shows the samples prepared with each mixture.

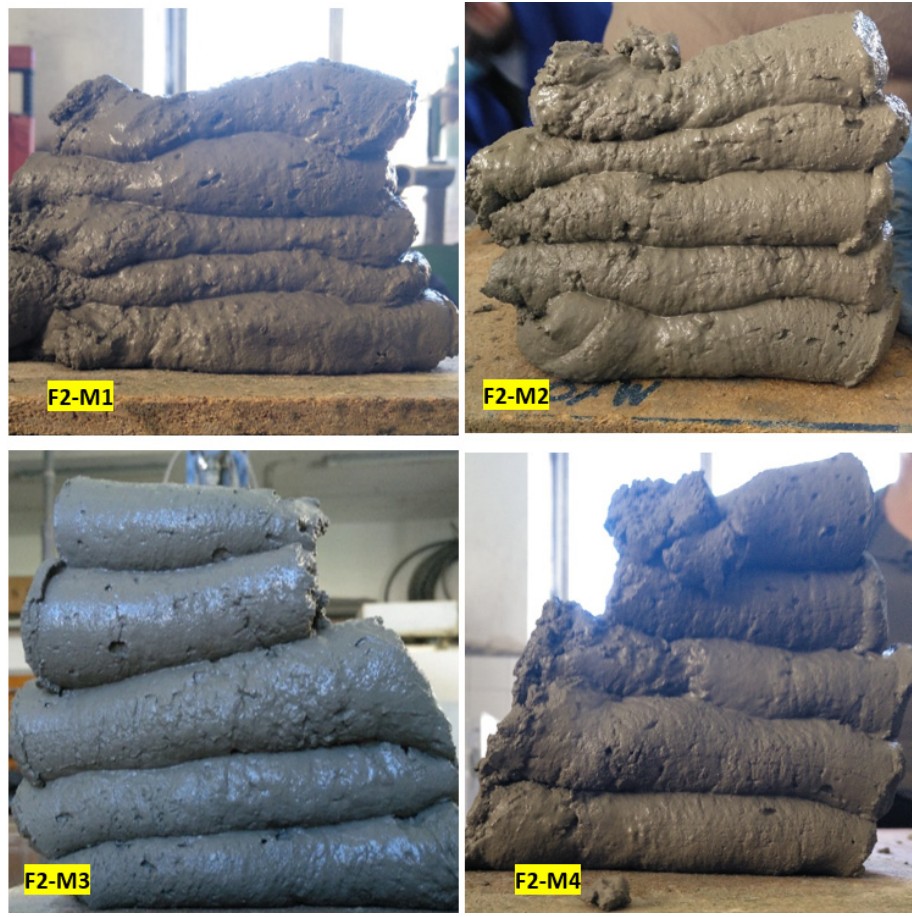

**Figure 6.** Samples manufactured by additive 3D printing with laboratory printing carriage.

From a visual observation, F2-M1 mixture is too fluid, and the layers are deformed too much. On the other hand, the F2-M4 mixture is too dry and does not allow the continuous pumping of material, which results in gaps and pumping failures. The F2-M2 and F2-M3 have the best behaviour. In the case of the F2-M2, it can be seen that there was a failure in the pumping of the fourth filament (Figure 7), which made the final section of the filament less thick. This failure, that only happened once, was due to a mechanical wear of the pump due to a larger particle, so the pump had to be repaired. As a consequence of the failure, the deposition of the next layer occurred irregularly.

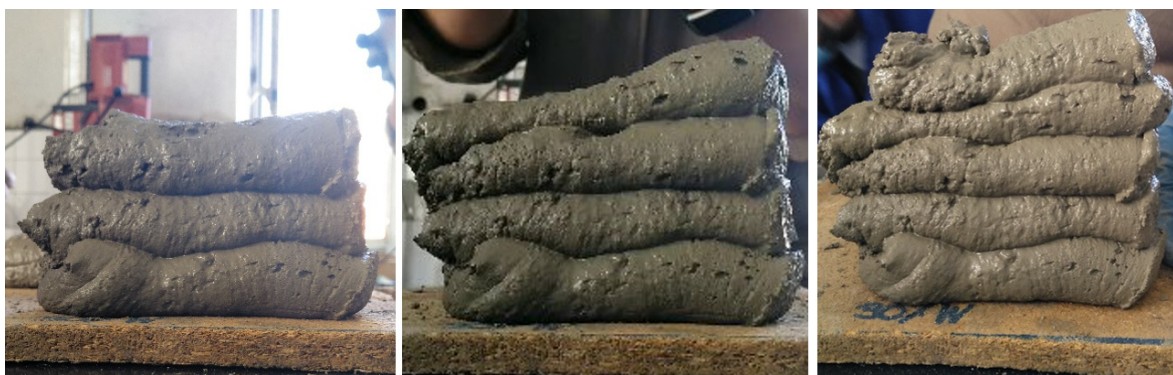

**Figure 7.** Pumping sequence of the F2-M2 mixture with failure in the fourth filament.

The next step was to check the influence of the manufacturing process on the strength of the printed material. Building using stacked filaments generates a discontinuous surface between them that can lead to weakness. In this case, the adhesion between layers was evaluated using a not-normalised penetration test that is widely used by slate companies in northern Spain [19]. The pieces were opened or separated in favour of their discontinuities using a flat chisel, similar to the artisanal exfoliation of the ornamental slate plates (Figure 8). The chisel was inserted in all interlayer, starting from the one closest to the edge. This way, there was only one layer to detach. The test was also carried out on the seamless material, which is used as a reference to quantify the strength reduction due to the joint. After that, the average of the tests of each group was made. The results are shown in Table 6, where material adhesion refers to the exfoliation strength of the seamless material and joint adhesion to the exfoliation strength of the joints.

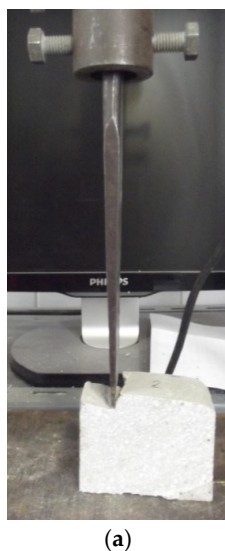 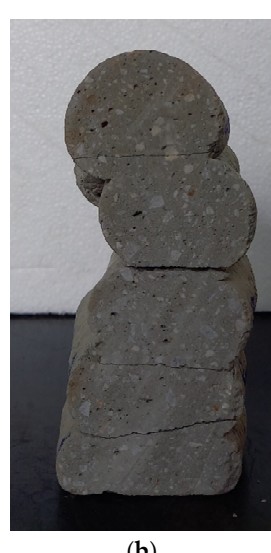 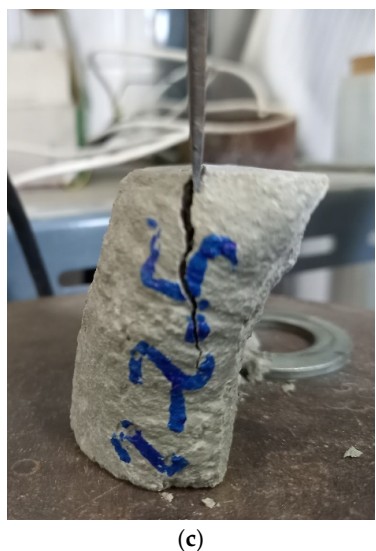

| (a) | (b) | (c) |

**Figure 8.** Chisel test: (**a**) device, (**b**) look of manufacturing joints, and (**c**) detail of one material test.

**Table 6.** Chisel adhesion test results.

| ID | Material Adhesion (MPa) | Joint Adhesion (MPa) | Strength Reduction (%) |
|---|---|---|---|
| F2-M1 | 2.06 | 1.37 | 33 |
| F2-M2 | 2.35 | 1.38 | 41 |
| F2-M3 | 1.74 | 1.67 | 4 |
| F2-M4 | 2.12 | 1.35 | 36 |

*4.4. Phase 4: Relationship between Fresh Strength and Buildability*

The tests described below were developed with two mixtures: F2-M2 and F2-M3. These two mixtures were chosen because they had the best behaviour in phase 3, and both of them have the same composition, with the exception of the accelerator. F2-M2 is without an accelerator, and F2-M3 is with an accelerator.

In order to determine the capacity of a filament to support the weight of the successive filaments deposited on it, the simple compressive strength was first considered as a parameter. As this was a fresh mixture, conventional methods were not suitable, so a handheld penetrometer, widely used in Soil Mechanics, was employed. This is a device that allows one to estimate the strength by the penetration of a stick attached to a foot of different diameters.

Figure 9 shows all the tests carried out on both mixtures. In the F2-M3 (with accelerator) the data series starts after 12 min, with strengths of 10 kPa, below which the equipment has no sensitivity. The F2-M2 (without accelerator) did not reach similar strength values until after one hour. After one and a half of setting (for mortars with an accelerator) or five hours (for the equivalent mortar without additives), the mortar was sufficiently hardened so that the handheld penetrometer could not be used. Therefore, in order to complete these tests, test-tubes were made for the simple compressive test, thus covering the entire time spectrum from the preparation of the mortar to four days of setting.

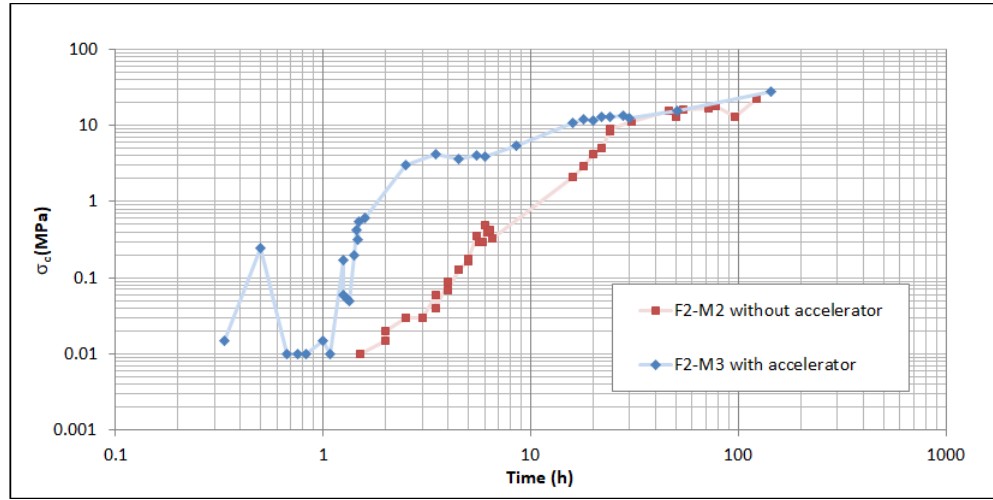

**Figure 9.** Comparison of compressive strength between mixtures F2-M2 (without accelerator) and F2-M3 (with accelerator).

As can be seen, the effect of the accelerator develops in the first two or three hours, when significant strengths are achieved that would speed up the manufacturing process. According to the results, the F2-M3 (with accelerator) must be used inside the first hour since its mixed, because this is the moment in which it begins to lose its workability. It is also clear that the handheld penetrometer is not suitable for evaluating the behaviour of the material below this time (or 1.5 h in the case of the mixture without accelerator).

In order to obtain the behaviour of the paste in those first moments, a device was designed consisting of a rod ending in a conical plastic nozzle and provided with a linear bearing and a 25 mm stroke sensor (see Figure 10). The sensor was connected to a data

acquisition card to measure the penetration-time curve, using a data acquisition frequency of 100 samples per second.

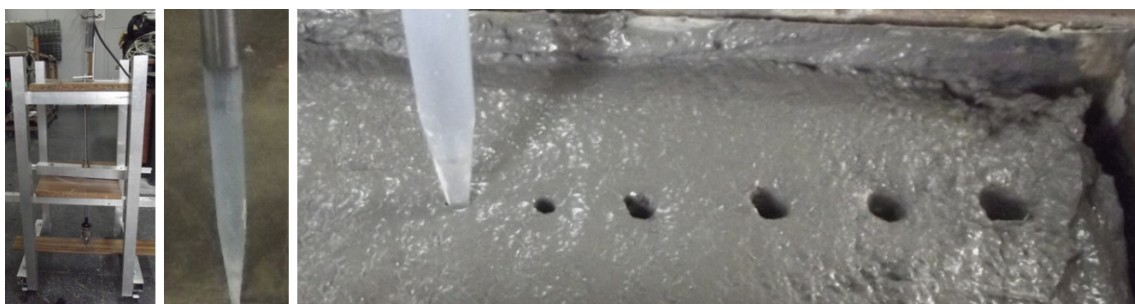

**Figure 10.** Penetration device.

By correlating penetration and strength in the section where data is available, a characteristic equation of the material is obtained. In the case of the mixture F2-M2 (without accelerator), this relationship is given by Equation (1), while for the mixture F2-M3 (with accelerator), Equation (2) is obtained:

$$Strength\ (kPa) = 94.8 \cdot e^{-0.30 \cdot Penetration\ (mm)} \tag{1}$$

$$Strength\ (kPa) = 591.7 \cdot e^{-0.14 \cdot Penetration\ (mm)} \tag{2}$$

From these relationships, the strength gain over time can be established. Figure 11 shows the relationships for both mixtures.

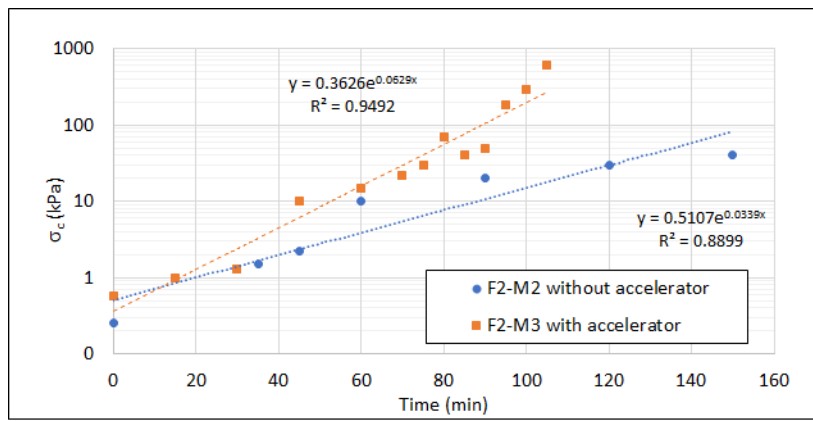

**Figure 11.** Strength gain of fresh mixtures.

## 5. Discussion

### 5.1. Experiment Results Summary

From phase 1 of the experiment, it can be concluded that in order to achieve materials with strength properties suitable for structural use with 3D concrete printing, no more than 20% of the aggregate can be replaced by recycled aggregate (that is to say, mining tailing from a flotation process with a high content of fine granulometry (fine sands, silts, and clays)) because the higher water consumption resulting from a higher specific surface area acts negatively on strength. Other less demanding applications with strength can admit higher percentages of these recycled materials.

From a strength standpoint, all mixtures designed during phase 2 have a single compressive strength in excess of the 25 MPa required for structural concrete. The addition of superplasticizer has two beneficial effects on strength. On the one hand, it allows the water/cement ratio to be lowered for the same consistency, and on the other, its own

chemistry improves the characteristics of the mixture (as can be seen by comparing, for the same water/cement ratio, the two mixes with different percentages of superplasticizer). These results are also in line with those of the adhesion of material (tests with a chisel on the material without manufacturing joints), although in this case the values are so close together that they would not be conclusive on their own.

With regard to the workability of the mixtures, the mini-cone and the flow table test indicate that the F2-M4 mixture is not fluid enough for the required application. With regard to the F2-M1 and F2-M2 mixtures, the behaviour seems contradictory in both tests. The mini-slump test confirms a more fluid consistency for the F2-M1 mixture, while in the flow table test the behaviour is opposite. Therefore, the addition of more superplasticizer allows a better shape retention under static conditions (mini-slump) but favours the flow under dynamic conditions (flow table test). This is appropriate for 3D printing, where flowability is required during pumping and a firm consistency after deposition. This change in the behaviour of the mixtures can be explained by the effect that non-ionic molecules combined with superplasticizers on the rheological parameters of mortar and concrete reducing the yield stress and plastic viscosity in mortar, while in concrete they only decrease the plastic viscosity [26–28].

This idea is corroborated during buildability tests. The greater fluidity of the F2-M1 mixture means that when some filaments are deposited on others, the deformation by its own weight is greater, not allowing the structure to maintain its shape and making successive overlapping difficult. In the case of the F2-M2 mixture, deposition takes place with equal fluidity, but the material, once pumped, maintains its shape to a greater extent, until a failure in the supply of the system generates a reduction in thickness and compromises the following layers. The F2-M3 mixture with the same composition of the F2-M2 but with an accelerator, improves its performance during printing and is the one with the best adhesion properties between layers. Finally, the F2-M4 mixture is very dry for application (coinciding with the results of the mini-slump and flow table test) and due to this one, it was not evaluated in phase 4.

### 5.2. Discussion

The development of the strength over time in the fresh mixtures F2-M2 (without accelerator) and F2-M3 (with accelerator) allows the evaluation of the construction possibilities of both mixtures. If the strength of the material is considered, the maximum weight that it would support with an acceptable deformation can be determined as a function of time. Considering the specific weight of the material to be 20 kN/m$^3$, the maximum height of material that would be supported by the mixture with accelerator F2-M2 as a function of time can be seen in Figure 12. It can be seen that a filament of 2.5 cm in height could be stacked every 7 min, which would allow a 1 m high element to be built in 4.6 h.

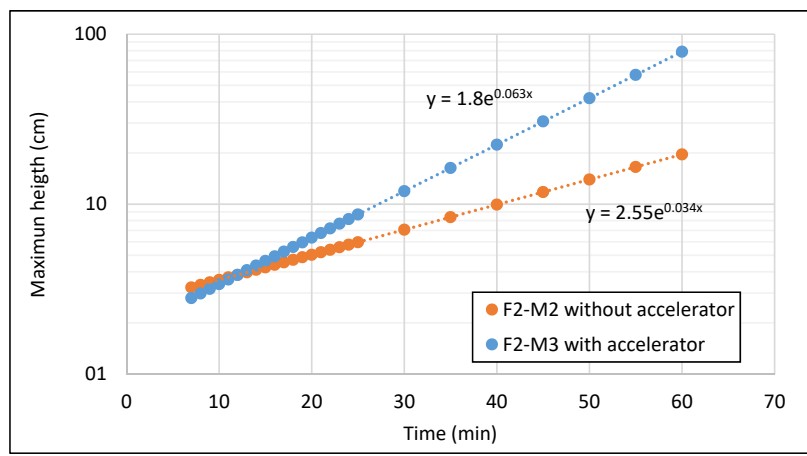

**Figure 12.** Height of material that is capable of supporting a filament as a function of time.

Considering that a printing robot could reach linear speeds of 5 m/min, in the seven minutes that should be left between layers, a 35 m long filament can be deposited. Therefore, in about 5 h, a 35 m long and 1 m high element could be built. Thus, with this type of material, a rectangular room of 10 m × 7.5 m × 2.5 m could be available in 12 h, reducing the time of a traditional construction.

## 6. Conclusions and Future Works

Throughout this research, the viability of using mining tailings as recycled aggregate has been demonstrated in an extremely demanding application, such as obtaining a material suitable for 3D concrete printing.

The requirements of strength, consistency, workability, and buildability were achieved by introducing up to 20% of this recycled material and adding a superplasticizer to the mixture that makes it possible to reduce the water/cement ratio to achieve a compromise between strength and workability. The buildability requirement, marked by the need to stack some layers of material on top of others, makes it necessary to use an accelerator to improve performance at an early age.

Strength tests on fresh mortars have shown that the rate of deposition of some filaments on others, marked by the rheology of the material, is perfectly compatible with the high speed of construction required for this construction process.

Although the results are promising, it is necessary to increase the number of tests carried out. In addition, a study of the recycled material used in each case must be made. The variability in properties such as granulometry or mineralogy among other properties produces different behaviour in the final mixture.

In the future, it will be important to study the ability to increase the percentage of mining tailing between 20–30%, optimize the ratio of superplasticizer and accelerator to improve the adhesion between layers, and carry out tests on a larger scale that allow verifying the limit number of stackable layers and its maximum straight length without the need to use supports or buttresses.

**Author Contributions:** Conceptualization, M.-I.Á.-F. and C.G.-N.; methodology, C.G.-N.; formal analysis, M.-I.Á.-F. and M.-B.P.-G.; lab tests, D.-J.G.-M. and J.E.M.-M.; writing—original draft preparation, D.-J.G.-M.; writing—review and editing, M.-I.Á.-F. and M.-B.P.-G.; supervision, C.G.-N. All authors have read and agreed to the published version of the manuscript.

**Funding:** This research was funded by "Gobierno del Principado de Asturias" within the Research Project grant number FC-GRUPIN-IDI/2018/000221.

**Institutional Review Board Statement:** Not applicable.

**Informed Consent Statement:** Informed consent was obtained from all subjects involved in the study.

**Data Availability Statement:** The data presented in this study are available on request from the corresponding author. The data are not publicly available due to confidentiality.

**Acknowledgments:** The authors want to acknowledge the support given by Spanish delegation of Master Builders Solutions, contributing the additives employed in the research.

**Conflicts of Interest:** The authors declare no conflict of interest. The funders had no role in the design of the study; in the collection, analyses, or interpretation of data; in the writing of the manuscript, or in the decision to publish the results.

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
