# Peer review of "Optimum Mix Design for 3D Concrete Printing Using Mining Tailings: A Case Study in Spain"

_sustainability, doi:10.3390/su13031568_

Round 1
Reviewer 1 Report
This paper aims to develop a mix with mine tailings for 3dcp. it can be helpful for the readers and get understanding on the material aspect of 3dcp. The reviewer has the following comments:
- Line 52, in ref. 6, “fresh properties for printing material were evaluated through tests with rheometers”. Was rheometer used in this paper? Please check
- Please provide the mix proportion for each mix. This will help the readers to understand.
- For the filler aggregate (sand), what is the maximum size? Is 6 mm too large? Please review more references and check the sand properties for most 3dcp mixes.
- Table 1. Is this the usual mixing process of printing concrete? please explain why the process is used. It is a bit complex.
- Some figures’ quantities should be improved.
- line 140, please provide all the test results if necessary.
- line 178, although the pumping system was described, more details are still needed. For example, how the concrete are pumped. Pump details? This is important.
- a circular nozzle was used in this study. To study the buildability of the mix, only 1 single bead was printed in each layer. Will this affect the stability of the 4-layer part?
- line 200. In the case of the F2-M2, it can be seen that there is a failure in the pumping of the fourth filament (Figure 8). Please explain.
- table 6, can you describe which layer the chisel was inserted? the interlayer properties might be different if the location was selected differently.
- Line 220. Does the compressive strength relates to the buildability?
- Fig. 10 needs improvement.
- addition of superplasticizer allows for better shape retention under static conditions (mini-slump) but favours flow under dynamic conditions (flow table test). Proofs are needed.
- is the concrete has a while colour? Reasons for this?
Author Response
The authors would like to thank you in advance for your comments.
- Line 52, in ref. 6, “fresh properties for printing material were evaluated through tests with rheometers”. Was rheometer used in this paper? Please check
Sorry for the confusion. The goal of using this reference is to introduce the use of more friendly tests as the tests employed in the presented research.
The sentence has been re-written as follow: In [6], fresh properties for printing material were evaluated through rheology tests.
- Please provide the mix proportion for each mix. This will help the readers to understand.
Thank you for your comment. The different mixes are in table 2 and table 3. In order to facilitate the reading, two new references have been added in line 101 and line 107.
- For the filler aggregate (sand), what is the maximum size? Is 6 mm too large? Please review more references and check the sand properties for most 3dcp mixes.
In sub-section 2.1 called Material it is indicated that the mix has fine sand with more than 99% less than 0.25 mm and an average size of 0.1 mm.
- Table 1. Is this the usual mixing process of printing concrete? please explain why the process is used. It is a bit complex.
There are several studios about the mixing process of printing concrete for example (Hermes LJ, Strength development of concrete used for 3D concrete printing determination of the influence of temperature on the development of concrete strength properties before initial set and the applicability of maturity methods, Doctoral thesis. Eindhoven University of Technology, 2018; Mennatallah AA, Effect of mixing water temperature on concrete properties in hot weather conditions, Doctoral thesis. The American University in Cairo, The School of Sciences and Engineering, 2013). In all of them, the focus is the water temperature and the mixing times, so in this paper has been followed a process that avoid the influence of these parameters.
These two references have been added to the paper and the rest of the references have been re-numbered.
- Some figures’ quantities should be improved.
All figures have been reviewed.
- line 140, please provide all the test results if necessary.
Thank you for your comment. The sentences line 140, now 141 has been re-written with the end to clarify why the other dosages have not been shown: “Although numerous dosages were checked many of them turned out to be unworkable. Table 2 shows the results of the most significant dosages as average value from three samples”.
- line 178, although the pumping system was described, more details are still needed. For example, how the concrete are pumped. Pump details? This is important.
Thank you for your advice. One sentence has been added at the end of the paragraph, line 192: “the pump is a laboratory gear pump with a flow rate of 12 l/min and a power of 0.3 kW.”
- A circular nozzle was used in this study. To study the buildability of the mix, only 1 single bead was printed in each layer. Will this affect the stability of the 4-layer part?
The authors fully agree with the reviewer. However, they have taken the criterion to use only one single bead because it is the most unfavorable situation. With more beads, corners or buttresses, buildability improves.
- line 200. In the case of the F2-M2, it can be seen that there is a failure in the pumping of the fourth filament (Figure 8). Please explain.
Thank you for the advice. There was a mechanical wear problem on the pump due to a particle with a larger diameter than due, so it had to be repaired. A new sentence has been added to the text: “This failure was due to a mechanical wear of the pump due to a larger particle, so the pump had to be repaired.”
- table 6, can you describe which layer the chisel was inserted? the interlayer properties might be different if the location was selected differently.
It has been inserted in all interlayer, starting from the one closest to the edge. This way, there is only one layer to detach. Subsequently, the average of the tests of each group is made.
The end of the paragraph has been re-written: “The chisel is inserted in all interlayer, starting from the one closest to the edge. This way, there is only one layer to detach. The test is also carried out on the seamless material, which is used as a reference to quantify the strength reduction due to the joint. After that, the average of the tests of each group is made. The results are shown in Table 6.”
- Line 220. Does the compressive strength relates to the buildability?
Yes, in the sense that each layer must support the weight of the upper layers without significantly deforming.
- 10 needs improvement.
Sorry for the mistake. The figure 10 has been corrected.
- addition of superplasticizer allows for better shape retention under static conditions (mini-slump) but favours flow under dynamic conditions (flow table test). Proofs are needed.
Although more tests are needed, the results show that along the deposition of successive layers, there is just no deformation in the lower layers due to the effect of the weight of successive layers. However, in the flow table test, these samples with superplasticizer open more, that is, they have a larger diameter. This indicates that under dynamic conditions they have less viscosity than those without a superplasticizer. This is important if piston pumps are used, and is intended to promote pumpability.
- is the concrete has a while colour? Reasons for this?
The authors feel the confusion. It is an effect of the lighting of the humid chamber.

Reviewer 2 Report
This work deals with mine tailings as a supplementary material for 3D printing of cementitious mixtures and investigates the effect of different fractions of mine tailings on strength, workability and printability in the presence and absence of superplasticizer/accelerator. At the end, it was concluded that a mixture with 20% of recycled material can be used as a printing mixture with satisfactory properties during printing and after curing.
There are major issues in this work especially in reproducibility and reliability of results and should be addressed before accepting this work for publication.
1) Line 100; What kind of mixer was used to prepare mixtures? Does it meet ASTM standards (e.g., ASTM C 305)?
2) Line 101; what are the test conditions and specimen dimensions for compressive strength test? Does it meet the ASTM standards like ASTM C-109?
3) Line 119-133; mini slump is normally utilized as flow test to measure diagonals after spreading of highly flowable mixtures (e.g., cement paste with superplasticizer). This method cannot lead to representative results for slump loss due to small quantity of the mixture, presence of large aggregates, lack of accuracy to measure loss, and more importantly user-dependent nature of the test and high probability of asymmetrical pull up. The latter is clearly visible in Figure 4 (F2-M1 10 and 20 min, F2-M2 all samples, F2-M3 1 and 20 min).
4) Line 138; three specimens for compressive strength of cementitious materials are not enough and don’t meet ASTM standards. The standard value is 6-10 samples. The average value and standard deviation should be reported.
5) Table 4, 5, and 6; Average value of at least 3 tests should be reported. Single point cannot be representing the behavior of a sample. If these results are average values, please report the number of the tests and standard deviation value.
6) Line 268-274; a contradictory behavior is reported here and the discussion is not informative. This kind of behavior is attributed to yield stress and plastic viscosity of a mixture and should be addressed to clarify the underlying mechanism.
7) Line 284-287; sample F4-M1 shows the best shape retention and highest quality of printing, however, no compressive strength, evolution of compressive strength, Chisel test, and workability results were provided.
Author Response
The authors would like to thank you in advance for your comments.
1) Line 100; What kind of mixer was used to prepare mixtures? Does it meet ASTM standards (e.g., ASTM C 305)?
For the kneading and taking into account the size of the granulometry of the materials (fine) a double sigma mixer has been used. One sentence has been added to the paragraph to clarify the doubt (line 100): “For the kneading and taking into account the size of the granulometry of the materials, a double sigma mixer has been used”
2) Line 101; what are the test conditions and specimen dimensions for compressive strength test? Does it meet the ASTM standards like ASTM C-109?
The last sentence of the paragraph has been re-written including: “… according to the UNE-EN 12390-3:2009 Standard”
3) Line 119-133; mini slump is normally utilized as flow test to measure diagonals after spreading of highly flowable mixtures (e.g., cement paste with superplasticizer). This method cannot lead to representative results for slump loss due to small quantity of the mixture, presence of large aggregates, lack of accuracy to measure loss, and more importantly user-dependent nature of the test and high probability of asymmetrical pull up. The latter is clearly visible in Figure 4 (F2-M1 10 and 20 min, F2-M2 all samples, F2-M3 1 and 20 min).
The authors agree with the comment. However, in this case and due to the granulometry of the material, the use of the Mini Slump is allowed. It is also necessary to indicate that all the tests were carried out by the same person, allowing the comparative analysis between the different mixtures tested under identical conditions. Finally, clarify that the test has been used as a qualitative and comparative test rather than a quantitative one.
At the end of the line 111 has been added the following comment: “because the granulometry of the material and …”
4) Line 138; three specimens for compressive strength of cementitious materials are not enough and don’t meet ASTM standards. The standard value is 6-10 samples. The average value and standard deviation should be reported.
The authors agree with the comment. But in this case the tests have the end of looking for a tendency and then preparing the new dosage. For that, three or five tests are enough. Once the dosage is optimised, and with the end of its industrial implementation, it is necessary increase the number of tests according to the standards. To clarify this concept, one sentence has been added at the end of the paragraph (line 142): “Although this number of tests is less than the standard says, the goal of the study is looking for the optimised dosage and then increase the number of test with the idea of the industrial implementation.”
5) Table 4, 5, and 6; Average value of at least 3 tests should be reported. Single point cannot be representing the behavior of a sample. If these results are average values, please report the number of the tests and standard deviation value.
All the tables show the average value of 3 samples. In table 2 and table 3. the deviations standard of the results have been included and the beginning of the paragraph with reference to table 2 has been modified: “Although numerous dosages were checked many of them turned out to be unworkable. Table 2 shows the results of the most significant dosages as average value from three samples.”. After the reference to table 3, it has added the sentence: “The results are the average value from three samples.”
6) Line 268-274; a contradictory behavior is reported here and the discussion is not informative. This kind of behavior is attributed to yield stress and plastic viscosity of a mixture and should be addressed to clarify the underlying mechanism.
More research is needed in this phenomenon, but it is a line to work on in the near future. At present, there is a great ignorance of the behavior of the mixes with superplasticizer, especially when the granulometry of the aggregate is so small [17]. To clarify the point, a sentence has been added at the end of the paragraph (line 285): “This contradictory behaviour shows the necessity of researching this phenomenon because there is a great ignorance of the behavior of the mixes with superplasticizer, especially when the granulometry of the aggregate is so small [17]”.
7) Line 284-287; sample F4-M1 shows the best shape retention and highest quality of printing, however, no compressive strength, evolution of compressive strength, Chisel test, and workability results were provided.
The authors agree with the reviewer. But once a good workability has achieved, a new study begins. In it, the time between layer will be optimize with the end of the accelerator does not affect the adherence between layers.

Round 2
Reviewer 1 Report
the comments have been suitably addressed.
Author Response
Thank you very much for your comments. English language and style has been revised again.
Reviewer 2 Report
The questions regarding standards and number of samples were addressed; however, I’m not convinced that authors put enough time and effort to address comments #6 and #7.
1- regarding comment 6; Authors stated that “there is a great ignorance of the behavior of the mixes with superplasticizer”. I refer the authors to look into published research of Professor Johann Plank from Technische Universität München and Karen Scrivener from Ecole Polytechnique Fédérale de Lausanne as two examples. The reported behavior in line 280-283, Page 11 can be explained through literature review.
2- Regarding comment 7: Authors responded as “But once a good workability has achieved, a new study begins” and based on that they legitimized the absence of data for the best performing sample. It’s not acceptable. There will be a similar ambiguity for readers as there is no data on the best performing sample.
3-Reference #1 should be replaced. Wikipedia is not a peer reviewed source.
Author Response
Dear reviewer, these are our responses to your comments:
1- regarding comment 6; Authors stated that “there is a great ignorance of the behavior of the mixes with superplasticizer”. I refer the authors to look into published research of Professor Johann Plank from Technische Universität München and Karen Scrivener from Ecole Polytechnique Fédérale de Lausanne as two examples. The reported behavior in line 280-283, Page 11 can be explained through literature review.
Thank you very much for your suggestion. The authors have analyzed the papers from Johann Plank and Karen Scrivener and the reported The reported behavior in line 280-283 has been explained with two references [18], [19] and the end of the paragraph has been re-written: “This contradictory behaviour can be explained by the effect that non-ionic molecules combined with superplasticizers on the rheological parameters of mortar and concrete reducing the yield stress and plastic viscosity in mortar, while in concrete they only decrease the plastic viscosity [17 -19].”
2- Regarding comment 7: Authors responded as “But once a good workability has achieved, a new study begins” and based on that they legitimized the absence of data for the best performing sample. It’s not acceptable. There will be a similar ambiguity for readers as there is no data on the best performing sample.
Thank you for your advice. The figure 13 and the previous paragraph has been deleted because the new dosage is being actually analyzed. The rest of the figures has been re-numbered.
3-Reference #1 should be replaced. Wikipedia is not a peer reviewed source.
The reference has been replaced: Silva Rotta, L.H., Alcântara E., Park E., Galante Negri R., Lin Y.N., Bernardo N., Gonçalves Mendes, T.S., Souza Filho C.R. The 2019 Brumadinho tailings dam collapse: Possible cause and impacts of the worst human and environmental disaster in Brazil. International Journal of Applied Earth Observation and Geoinformation 2020, 90.

Round 3
Reviewer 2 Report
All comments have been addressed in this version; no new comments.
Author Response
Thank you very much